# Revealing Prognostic and Immunotherapy-Sensitive Characteristics of a Novel Cuproptosis-Related LncRNA Model in Hepatocellular Carcinoma Patients by Genomic Analysis

**DOI:** 10.3390/cancers15020544

**Published:** 2023-01-16

**Authors:** Zhenzhen Mao, Ye Nie, Weili Jia, Yanfang Wang, Jianhui Li, Tianchen Zhang, Xinjun Lei, Wen Shi, Wenjie Song, Xiao Zhang

**Affiliations:** 1Xi’an Medical University, Xi’an 710021, China; 2Department of Hepatobiliary Surgery, Xijing Hospital, Fourth Military Medical University, Xi’an 710032, China; 3The State Key Laboratory of Cancer Biology, Department of Biochemistry and Molecular Biology, Fourth Military Medical University, Xi’an 710032, China; 4Research Office of the Institute of Tropical Medicine, Hainan Hospital of PLA General Hospital, Sanya 572013, China

**Keywords:** cuproptosis-related LncRNAs (crLncRNAs), immunotherapy, tumor immune microenvironment (TIME), immune checkpoint inhibitors (ICIs), AL365361.1

## Abstract

**Simple Summary:**

Hepatocellular carcinoma (HCC) remains a major health concern. Immunotherapy combined with targeted therapy brings hope to patients with HCC, but its primary beneficiaries have not been identified. Recent studies have found that copper induces cell death, which is named cuproptosis. As we know, cell death is closely related to tumor therapy, in which some non-coding RNAs involved. Therefore, we focused on whether some long non-coding RNAs (LncRNAs) are related to cuproptosis, and the cuproptosis-related LncRNAs (crLncRNAs) can classify tumor treatment-sensitive populations. In the study, we explore a model of crLncRNAs with excellent specificity and sensitivity that is capable of predicting the prognosis of HCC patients and classifying tumor immunotherapy-sensitive populations, thereby providing new insights for the development of appropriate clinical strategies.

**Abstract:**

Immunotherapy has shown strong anti-tumor activity in a subset of patients. However, many patients do not benefit from the treatment, and there is no effective method to identify sensitive immunotherapy patients. Cuproptosis as a non-apoptotic programmed cell death caused by excess copper, whether it is related to tumor immunity has attracted our attention. In the study, we constructed the prognostic model of 9 cuproptosis-related LncRNAs (crLncRNAs) and assessed its predictive capability, preliminarily explored the potential mechanism causing treatment sensitivity difference between the high-/low-risk group. Our results revealed that the risk score was more effective than traditional clinical features in predicting the survival of HCC patients (AUC = 0.828). The low-risk group had more infiltration of immune cells (B cells, CD8^+^ T cells, CD4^+^ T cells), mainly with anti-tumor immune function (*p* < 0.05). It showed higher sensitivity to immune checkpoint inhibitors (ICIs) treatment (*p* < 0.001) which may exert the effect through the AL365361.1/hsa-miR-17-5p/NLRP3 axis. In addition, NLRP3 mutation-sensitive drugs (VNLG/124, sunitinib, linifanib) may have better clinical benefits in the high-risk group. All in all, the crLncRNAs model has excellent specificity and sensitivity, which can be used for classifying the therapy-sensitive population and predicting the prognosis of HCC patients.

## 1. Introduction

Liver cancer is the sixth most common cancer worldwide and the third leading cause of cancer death [1]. Hepatocellular carcinoma (HCC) is the most common type of liver cancer, accounting for approximately 80% of cases, and its 5-year survival rate is less than 20% [2]. Chronic hepatitis B virus infection is the main pathogenic factor for HCC in China. After chronic hepatitis progresses to cirrhosis, about 85% of patients with cirrhosis are diagnosed with HCC [3]. Early diagnosis of HCC is difficult, with two-thirds of patients being unable to undergo radical surgery, and for remaining one-third, the recurrence rates are high, and the survival rate is low. Targeted therapies have emerged, but with low treatment response rate and high drug resistance [4]. The increase in drug toxicity and side effects aggravates the poor prognosis of patients with advanced HCC. Immune checkpoint inhibitors (ICIs) are promising therapies for HCC based on early efficacy data. Therefore, the pembrolizumab monotherapy [5], the combination of ezetimibe and bevacizumab [6], and nivolumab plus ipilimumab combination [7] have received the US Food and Drug Administration (FDA) approval successively. Although the application of ICIs has improved the prognosis of HCC patients who showed a good response [8], the response rate is less than 30% [9]. Biomarkers such as tumor immune microenvironment (TIME) [10], tumor mutation load (TMB) [11], immune checkpoint genes [12], immune score, and IPS score [13] may predict sensitivity to ICI treatment, providing the possibility of identifying patients who are sensitive to ICI treatment, but this approach still lacks specificity and accuracy. Immunotherapy combined with targeted therapy has a low response rate and high drug resistance. Therefore, it is particularly important to identify susceptible individuals [14,15]. Given the limited therapeutic strategies for HCC, new prognostic models need to be developed for prognosis and to identify sensitive treatment options, which may improve the survival rate of patients with HCC by guiding treatment decisions.

Long non-coding RNAs (LncRNAs) are one kind of non-coding RNA with more than 200 nucleotides in length, which play a variety of roles in regulating immune response and affecting tumor progression [16]. For example, LncRNA MIR155HG upregulates the expression of PD-L1 through the miR-223-STAT1 axis and promotes the immune escape of HCC [17]. Previous studies have found that LncRNA is involved in the regulation of pyroptosis, ferroptosis, and other common programmed cell death modes during tumor progression, so it can be used as a biomarker to predict the prognosis of cancer patients and the response to immunotherapy. For instance, pyroptosis-related LncRNAs (HPN-AS1, MED8-AS1, SREBF2-AS1, MKLN1-AS, and ZNF232-AS1) can predict the response to immunotherapy in HCC patients [18]. Similarly, ferroptosis-related LncRNAs (MKLN1-AS, LINC01224, LNCSRLR, LINC01063, PRRT3-AS1, and POLH-AS1) are reliable in predicting the prognosis and immunotherapy response in HCC patients [19]. Hence, we believe LncRNAs may play a significant role in multiple forms of cell death, and the recent discovery of cuproptosis has attracted our attention. Cuproptosis is a newly defined programmed cell death type [20]. Copper ions induce a proteotoxic stress response by binding to thioacylated tricarboxylic acid cycling-related enzymes that ultimately lead to cell death [21]. A disorder of copper metabolism can lead to the metabolites disrupting the tricarboxylic acid cycle. Copper ion levels are significantly changed in the tumor tissues of many cancer patients [22,23], and they affect tumor progression by affecting mitochondrial energy metabolism [24,25,26]. It has been demonstrated that blocking Cu^2+^ transport induces an ATP decrease, activates AMP-activated protein kinase, and ultimately inhibits the proliferation of tumor cells [27]. However, whether LncRNAs participate in the regulation of tumor cuproptosis has not been reported previously. Therefore, we hope to clarify the relationship between them and assess how the LncRNAs might affect tumor progression by regulating cuproptosis. In addition, we attempted to use the cuproptosis-related LncRNAs (crLncRNAs) as novel biomarkers for immunotherapy and targeted therapy in patients with hepatocellular carcinoma to establish a prognostic model.

In this study, we designed to provide a promising biomarker to predict the prognosis of HCC patients, the tumor microenvironment, classify the treatment-sensitive population, and provide new insights for developing appropriate clinical strategies.

## 2. Materials and Methods

### 2.1. Data Acquisition and Screening

We downloaded RNA-seq data and corresponding clinical characteristics from the Cancer Genome Atlas (TCGA) database (424 HCC samples, including 50 normal samples and 374 tumor samples), which updated its data after April 2022 (https://portal.gdc.cancer.gov/, accessed on 31 May 2022). Considering the possibility of non-cancer death, HCC patients whose survival time was <30 days or uncertain were excluded. Subsequently, data from 343 patients were analyzed. The baseline characteristics of patients are summarized in Appendix A. TCGA is a public database and does not require ethical approval.

### 2.2. Identification of CrLncRNAs

A validated list of genes (NFE2L2, NLRP3, ATP7B, ATP7A, SLC31A1, FDX1, LIAS, LIPT1, LIPT2, DLD, DLAT, PDHA1, PDHB, MTF1, GLS, CDKN2A, DBT, GCSH, and DLST) associated with cuproptosis was obtained from a recent study by Tsvetkov et al. [28]. Spearman correlation coefficients were calculated based on cuproptosis-related mRNA and LncRNA expression profiles to identify crLncRNAs (|R| > 0.4 and *p* < 0.001).

### 2.3. Construction of the Prognostic Model of CrLncRNAs

The crLncRNAs correlated with HCC survival time were determined by univariate Cox regression analysis (*p* < 0.05). The 343 patients were randomly assigned to either the training cohort or the test cohort in a ratio of 3:2. Appendix A summarizes the baseline characteristics of the two groups. For the training cohort, LASSO regression analysis was performed using the R project ‘Glmnet’ package. Parameter selection was adjusted by a round of cross-validation to prevent overfitting, and the partial likelihood deviation met the minimum criterion. Then, the multivariate Cox regression was performed on the 9-crLncRNA generated, and the corresponding coefficients were multiplied to obtain the score of each sample. The formula was determined as follows: Risk score = AC026412.3 * 2.54199 + AC026356.1 * 1.87356 + SLC6A1-AS1 * (−0.71370) + AC011462.4 * 1.19584 + MIR548XHG * 0.62675 + AL031985.3 * 0.96845 + AL117336.2 * 0.77657 + MCM3AP-AS1 * 2.69411 + AL365361.1 * (−2.99107).

### 2.4. Confirmation of the Prognostic Signatures of 9-CrLncRNA

Patients in the training and validation cohorts were divided into high- and low-risk groups according to the median risk score. Principal component analysis (PCA) of the 9-crLncRNA was performed by the ‘scatterplot3d’ R package. The Kaplan–Meier survival analysis was conducted using the ‘survminer’ R package, with the log-rank test comparing the overall survival (OS) and progression-free survival (PFS) between high- and low-risk groups. ROC curves were constructed to verify the predictive power of features, and the area under the ROC curves (AUC) values for 1, 3, and 5 years were calculated by the ‘timeROC’ R package. Risk curves, survival status, and heatmaps of risk gene expression profiles were generated by the ‘pheatmap’ R package. In addition, univariate and multivariate Cox regression analyses were performed, and a conformance index (C-index) was used to assess whether the risk score could be an independent predictor of OS in HCC patients.

### 2.5. Nomogram Construction and Assessment

The R packages ‘regplot’ and ‘rms’ were used to construct the nomogram of the risk score, and the prognosis of HCC patients in it at years 1, 3, and 5 were estimated. Finally, the ROC curve, decision curve analysis (DCA) curve, and calibration curve were used to evaluate the nomogram’s accuracy and reliability.

### 2.6. Enrichment Analysis

High- and low-risk groups of differentially expressed genes (DEGs) were filtered through ‘limma’ R package (log2 |FC| > 1, FDR < 0.05). Gene Ontology (GO) functional analysis and Kyoto Encyclopedia of Genes and Genomes (KEGG) pathway analysis were performed using the ‘clusterProfiler’ R package (FDR < 0.05, *p* < 0.05).

### 2.7. Evaluation of Immune Cell Infiltration and Immune Microenvironment

CIBERSORT is an analysis tool that transforms expression data into the absolute abundance of immune and stromal cell expression profiles, assessing the proportion of 22 human immune cell subpopulations (http://CIBERSORT.stanford.edu/, accessed on 31 May 2022). The CIBERSORT algorithms were used to quantify the relative infiltration levels of different immune cell subsets. ESTIMATION algorithms were used to calculate the immune/estimate/stromal scores and the purity of tumor in HCC patients [29,30]. Finally, we used the single sample gene set enrichment analysis (ssGSEA) method to analyze immune-related functions and immune cell infiltration profiles between the high-risk and low-risk groups.

### 2.8. Assessment of the Options of Specific ICIs Treatment

The immunophenoscore (IPS) of HCC patients was obtained from the Cancer Immunome Atlas (https://tcia.at/, accessed on 31 May 2022) to compare the potential utilization of immunotherapy in the high- and low-risk subgroups. Immune checkpoint gene expression is associated with ICIs response, so we analyzed differences in immune checkpoint gene expression between the two subgroups. In addition, Spearman correlation analysis identified 9 crLncRNAs that are most closely associated with immune checkpoints.

### 2.9. GSVA Analysis

GSVA analysis boosts the ability to detect subtle changes in pathway activity in sample populations. The ‘GSEABase’ and ‘GSVA’ R packages were used to analyze the correlation between the KEGG pathway and the prognostic signatures of 9 crLncRNAs, paving the way for the construction of pathway-based biological models.

### 2.10. Prediction of Targeted MiRNAs

We predicted microRNA targets through TargetScan (https://www.targetscan.org/vert_71/, accessed on 18 October 2022). The miRNA–LncRNA relationship was predicted through DIANA Tools (https://diana.e-ce.uth.gr/lncbasev3/interactions, accessed on 18 October 2022). Some diagrams were used online graphic drawing sites (https://app.rawgraphs.io/, accessed on 18 October 2022) (https://www.omicstudio.cn/tool, accessed on 18 October 2022). Based on the online Venn diagram tool, we screened two common genes (https://bioinfogp.cnb.csic.es/tools/venny/index.html, accessed on 18 October 2022).

### 2.11. Copy Number Variation (CNV) Analysis

Statistical data of heterozygous and homozygous CNV mutation were displayed as pie charts, and the gene expression significantly affected by CNV was obtained via the Pearson correlation analysis. (http://bioinfo.life.hust.edu.cn/web/GSCALite, accessed on 22 June 2022).

### 2.12. Drug Sensitivity Analysis

The gene set drug resistance analysis from Genomics of Drug Sensitivity in Cancer (GDSC) IC50 drug data. The Spearman correlation analysis was used to determine the drug sensitivity and gene expression profile data of tumor cell lines. (http://bioinfo.life.hust.edu.cn/web/GSCALite, accessed on 22 June 2022).

### 2.13. HCC Tissue Specimens

With the informed consent of HCC patients in Xijing Hospital, 10 pairs of fresh frozen tumors and adjacent normal tissues were collected. These patients did not receive any treatment before surgery. This research was approved by the Medical Ethics Committee of the First Military Medical University.

### 2.14. Real-Time Quantitative PCR Analysis

TRIzol reagent (Invitrogen, Carlsbad, CA, USA) was used to extract total RNA from fresh HCC tissues from Xijing Hospital according to the instructions. 1 μg of RNA was employed to synthesize cDNA using the PrimeScript RT Reagent Kit Perfect Real Time (TaKaRa, Dalian, China) or the miScript II RT Kit (Qiagen, Hilden, Germany). In order to detect the mRNA and LncRNA levels of target genes, fluorescent qRT-PCR was used, with GAPDH as the internal control. The expression level of miR-17-5p in HCC tissues was detected by qRT-PCR, and the expression of U6 snRNA (sn-RNU6) was used as the internal control. All the primers were obtained from AuGCT (Beijing, China), and the reactions were repeated three times. The relative expression levels of LncRNA, miRNA, or mRNA were obtained and analyzed using the Bio-Rad C1000 thermal cycling apparatus (Bio-Rad, Hercules, CA, USA). The primer sequences are listed in Appendix A.

### 2.15. Statistical Analysis

We used R version 4.2.0 to statistically analyze our data and graph visualization. The Kaplan–Meier curve was adopted to plot the prognostic survival curve for the subgroups, and the log-rank test was performed to evaluate if the differences in OS were statistically significant. The Spearman method was used to calculate the correlation between two variables. The differences in the proportions of clinical characteristics were analyzed by the chi-squared test. The Wilcoxon test was used for the analysis of differences between the two independent groups. *p* < 0.05 was considered statistically significant.

## 3. Results

### 3.1. Construction and Validation of 9-CrLncRNA Prognostic Model in HCC

The flow chart of our study is shown in Figure 1. The TCGA transcriptome data included 50 normal cases and 374 HCC patients. A total of 16,876 LncRNAs, 19,938 mRNAs, and 19 cuproptosis-related genes’ transcriptome data were obtained from TCGA. The co-expression network of mRNAs and crLncRNAs is plotted in Figure 2A. Finally, a total of 394 crLncRNAs were obtained by Spearman correlation analysis, and 147 of them are closely related to patient survival. The cvfit and lambda curves are shown in Figure 2B,C. The prognostic risk model was based on 9 crLncRNAs (AC026412.3, AC026356.1, MCM3AP-AS1, AL031985.3, AL117336.2, AL365361.1, SLC6A1-AS1, MIR548XHG, AC011462.4). The cuproptosis-related mRNAs and the 9-crLncRNA signatures were significantly correlated, among which, the 9-crLncRNA signatures significantly correlated with GLS, LIPT1, MTF1, ATP7A, and NLRP3 (Figure 2D).

In this model, the following formula was used to calculate the risk score for each HCC patient in the TCGA database: Risk score = AC026412.3 * 2.54199 + AC026356.1 * 1.87356 + SLC6A1-AS1 * (−0.71370) + AC011462.4 * 1.19584 + MIR548XHG * 0.62675 + AL031985.3 * 0.96845 + AL117336.2 * 0.77657 + MCM3AP-AS1 * 2.69411 + AL365361.1 * (−2.99107) (note: the name of LncRNA indicates their expression level in TCGA database).

The PCA showed that our prognostic prediction model could separate the two groups’ patients (*p* < 0.05, Figure 2E–H). We first validated the predictive ability of the prognostic model in HCC patients with different clinical characteristics. There were significant differences between G1 and G3 and grades G2 and G3, between stages I and II and III and IV, and between T1 and 2, and T3 and 4 (*p* < 0.05) (Figure 3A–C). Interestingly, the OS rate of HCC patients can be well predicted by the risk scores for age (<65, ≥65), sex (female, male), grade (G1-2, G3-4), stage (III–IV), stage T (T3-4), M0, and N0 (*p* < 0.05). And, within these clinical-characteristic groups, patients in the high-risk group had a worse prognosis when compared to the low-risk group (Figure 3D,I,K,L,N,O). However, patients in stages I–II and T1–2 had a better prognosis when compared to the low-risk group and may not be classified by risk scores (*p* > 0.05) (Figure 3J,M). These results indicate that the survival of patients with advanced HCC can be predicted based on risk scores in different clinical characteristic groups.

### 3.2. The 9-CrLncRNA Model Has High Reliability in Predicting the OS and Application Value in Clinical

Subsequently, we assessed the prognostic value of the 9-crLncRNA. According to the risk and survival status distributions of the visualized risk score, the sample distribution was reasonable. The up-regulated genes of AC026412.3, AC026356.1, MCM3AP-AS1, AL031985.3, AL117336.2, MIR548XHG, and AC011462.4 in the high-risk groups of the TCGA-HCC database were considered risk crLncRNAs. AL365361.1 and SLC6A1-AS1 were protective crLncRNAs, as they were downregulated in the high-risk group (blue: low expression level; red: high expression level, Figure 4A). The OS rate of HCC patients in the high-risk group was lower than that in the low-risk group (Figure 4D). In addition, the areas under the curve (AUC) in the training group were 0.836, 0.780, and 0.794 in 1, 3, and 5 years, respectively, indicating that features had a strong ability to predict survival time (Figure 4G). The multi-indicator ROC curve indicated that the risk score was significantly better than traditional clinical characteristics at predicting the OS rate (AUC = 0.836) (Figure 4J). To further evaluate the predictive validity of this 9-crLncRNA model, dual validation was performed in both the validation group and the entire group using distribution maps, heat maps, Kaplan–Meier survival curves, and ROC curves. Samples from the two risk groups were also reasonably distributed in the validation group (Figure 4B,E,H,K) and the entire group (Figure 4C,F,I,L). As expected, the high-risk group has a higher mortality rate than the low-risk group. All of the above results indicate that our model has a good specificity and sensitivity.

First, the univariate Cox regression analysis showed that the stage, T staging, and risk scores of the 9-crLncRNA characteristics were correlated with the OS rate (*p* < 0.001) (Figure 5A). Furthermore, risk score was an independent risk prognostic factor for predicting the OS rate (HR > 1, *p* < 0.001) (Figure 5B). Risk score concordance is higher than other clinical features, suggesting that the model could predict survival better than others (Figure 5C). The nomogram calculated the likelihood of survival for these patients by summing scores of clinical characteristics (Figure 5D). The calibration plot of 1-, 3-, and 5-year survival probabilities showed that the nomogram based on risk score had good predictive ability in clinical application (Figure 5E). Moreover, the multi-indicator ROC curves and DCA curves showed that while both nomogram (AUC = 0.805) and risk score were good predictors of OS, the latter was better (AUC = 0.824) (Figure 5F,G). Therefore, the risk score of our model has a high application value in clinical settings and is reliable in predicting the OS of HCC patients.

### 3.3. The 9-CrLncRNA May Be Closely Related to Tumor Immunity

We obtained 357 DEGs associated with cuproptosis, including 125 up-regulated genes and 132 down-regulated genes (Figure 6A). The KEGG pathway analysis indicated that cuproptosis-related DEGs were mainly enriched in the p53 signaling pathway (hsa04115), cell cycle (hsa04110), oocyte meiosis (hsa04114), etc. (Figure 6B). In the biological process category, GO analysis showed that DEGs were mainly enriched in immune-response related pathways, such as cell recognition, complement activation, humoral immune response mediated by circulating immunoglobulin, phagocytosis (recognition and engulfment), B cell receptor signaling pathway, positive regulation of B cell activation, etc. In the cellular component category, the DEGs were mainly enriched in immunoglobulin complex, etc. In the molecular function category, the DEGs were mainly enriched in antigen binding, immunoglobulin receptor binding, etc. (Figure 6C). All in all, the cuproptosis-related DEGs are mainly enriched in the cell cycle (hsa04110), antigen binding (GO:0034987) and immune response related pathway mediated by circulating immunoglobulin (GO:0042571) (Figure 6D,E). Therefore, the DEGs obtained according to our 9-crLncRNA signatures grouping were mainly related to tumor immunity, which is convenient for further study. We have reasons to believe that 9-crLncRNA may be closely related to tumor immunity.

### 3.4. The Low-Risk Group Was More Likely to Have a Higher Immune Response

To further explore the relationship between the prognostic signatures of 9-crLncRNA and tumor immunity in HCC patients, we used the CIBERSORT algorithm to determine the immune cell infiltration in all HCC patients from the TCGA database. We analyzed the stromal score (substrate cells in tumor tissue), immune score (infiltrating immune cells in tumor tissue), and estimated score (the sum of the stromal and immune scores from individual cases) in the high- and low-risk groups. A higher immune score indicates a better outcome, and as expected, the immune score was significantly higher in the low-risk group (*p* < 0.001, Figure 7A). We compared the differences in each immune cell between the low- and high-risk groups and found that CD8^+^ T cells and macrophage M0 were significantly different between the two groups (Figure 7B). Subsequently, the ssGSEA was applied to the RNA sequencing data of HCC samples to assess immune cell infiltration and related functions. Immune cell populations, including naive B cells, B cell memory, plasma cells, CD8^+^ T cells, CD4^+^ T cell memory resting, M0 macrophages, and dendritic cells activation, were found to be significantly different between the two groups (Figure 7C). The high-risk group had a poor prognosis, which was characterized by low ratio of B cell naive, resting B cell memory, plasma cells, CD8^+^ T cells and CD4^+^ T cells memory, and the high activation of M0 macrophages and dendritic cells. In addition, a comparison of immune characteristics between high- and low-risk patients showed that low-risk patients have higher APC_co-inhibition, APC_co-stimulation, chemokines and chemokine receptors (CCR), checkpoint, cytolytic activity, HLA, inflammatory promotion, parainflammation, T_cell_co-inhibition, and T_cell_co-stimulation, type-I IFN response, and type-II IFN response (Figure 7D). This means that patients in the low-risk group are more likely to have a higher immune response, and they can benefit more from immunotherapy. Finally, the expression level of each patient’s immune cells and immune cell-related functions between different risk groups is summarized in Figure 7E. Patients in the low-risk group had higher expression levels. It is consistent with the conclusion that patients in the low-risk group may have a higher immune response. We speculate that the 9-crLncRNA may change the TIME, one of the biomarkers representing sensitivity to ICIs treatment, by influencing immune cell infiltration. Therefore, the risk score based on 9-crLncRNA may be able to identify the population susceptible to ICIs treatment.

### 3.5. Patients in the Low-Risk Group May Have a Higher Sensitivity to ICIs

The TIME was different between the two risk groups in our study; therefore, we hypothesized that there may be differences in the efficacy of ICIs treatment. Due to the importance of ICI in the immunotherapy of hepatocellular carcinoma, we further analyzed the differential expression of immune checkpoint genes between the two groups, and found that the expression in the low-risk group is higher, while the results for TNFSF4 and CD276 are the opposite (Figure 8A). This means that patients in the low-risk group may benefit from ICIs treatment. Interestingly, the IPS, IPS_CTLA4 + PD1, IPS_CTLA4, and IPS_PD1 in low-risk groups had higher values (Figure 8B–E). This means that patients in the low-risk group may have higher ICIs sensitivity and benefit from immune checkpoint inhibitor therapy, whether they are treated with a CTLA-4 inhibitor alone, a PD-1 inhibitor, or both.

### 3.6. AL365361.1 May Be the Main Tumor Immune-Related Molecule of the 9-CrLncRNA Model

Our 9-crLncRNA model has excellent specificity and sensitivity to identify the sensitive population of immune checkpoints inhibitor treatment. Therefore, further analysis of the potential function of the 9 crLncRNAs may provide the possibility of finding valuable therapeutic targets. Figure 9A shows the potential pathways involved in the prognostic signatures of 9-crLncRNA analyzed by GSVA. It was found that 9-crLncRNA mainly mediated immune-related pathways (such as T cell receptor, B cell receptor, and JAK-STAT signaling pathway). Subsequently, we analyzed the correlation between 9 crLncRNAs and 22 immune cells and their immune-related functions (Figure 9B). AL365361.1 was significantly correlated with T cells, APC, immune checkpoint, and CCR (*p* < 0.001). As mentioned above, immune checkpoint genes were differentially expressed between the low-risk and high-risk groups, so we analyzed the association between the 9 crLncRNAs and immune checkpoint genes (Figure 9C). The data showed that AL365361.1 was significantly correlated with immune checkpoints CD28 and CD40LG (*p* < 0.001), which have been reported to synergistically regulate exhaustion of tumor-infiltrating lymphocytes (TILs) and response to PD-1 blockade [31,32]. This implies that AL365361.1 plays a tumor suppressive role in HCC patients. To confirm the possibility of this relationship, we collected fresh tissues and performed qRT-PCR to examine the expression level of AL365361.1 in HCC tissues and adjacent normal tissues from the HCC patients (Figure 9D). The result showed that the expression of AL365361.1 in HCC tissues is significantly lower than that in adjacent normal tissues. According to the above results, AL365361.1 may be the main tumor immune-related molecule between the two risk groups.

### 3.7. Sensitive Drugs for NLRP3 Mutation May Improve OS in High-Risk Group

AL36536.1 is closely related to tumor immunity and screened from cuproptosis-related genes. Whether it affects tumor immunity by mediating cuproptosis-related genes has triggered our thinking. At the same time, although our model was able to distinguish the sensitive population of immune checkpoints inhibitor therapy well, we hope to use it not only to distinguish the sensitive population but also to propose new treatment options for patients in the high-risk group who are not sensitive to ICIs, thereby prolonging the OS of the high-risk group. Therefore, we analyzed the correlation between 9 cuproptosis-related genes and 22 immune cells and their immune-related functions (Figure 10A). Interestingly, NLRP3 was also significantly associated with T cells, APC, immune checkpoints, and CCR (*p* < 0.001), like AL365361.1. Our previous results showed that AL365361.1 and NLRP3 are highly expressed in the low-risk group. This supports our previous findings that patients in the low-risk group may be sensitive to ICIs.

We found that NLRP3 may be an important target for improving OS in high-risk patients. The proportion of NLRP3 activation/inhibition pathways was analyzed and visualized by GSCALite (Figure 10B). NLRP3 mainly activates epithelial-to-mesenchymal transition (EMT) and hormone pathways. NLRP3 in HCC tumors and adjacent normal tissues were analyzed for copy number variation (CNV) (Figure 10C,D). NLRP3 had the most significant changes, mainly with CNV mutations (>60%). Furthermore, the expression of NLRP3 in HCC tissues is significantly lower than that in adjacent normal tissues (Figure 10E). Genomic resistance analysis showed that VNLG/124, sunitinib, and linifanib were sensitive drugs in the inflammatory body NLRP3 mutant population (Figure 10F). Our previous results showed that NLRP3 was poorly expressed in the high-risk group; it is dominated by NLRP3 mutation and insensitive to immune checkpoint inhibitors. Thus, in high-risk groups, NLRP3 mutation increased susceptibility to VNLG/124, sunitinib, and linifanib. This may extend the survival of high-risk group patients who are not sensitive to ICIs.

### 3.8. The Potential Regulation Axis between AL365361.1 and NLRP3

The risk score based on 9-crLncRNA can identify the sensitive population for ICIs treatment, and the mechanism involved in the regulation of core molecules has aroused our attention. Our previous results showed that AL365361.1 was positively correlated with NLRP3 (Figure 11A), and both of them were positively correlated with anti-tumor immunity. In conclusion, we speculated that NLRP3 may cooperate with AL365361.1 to regulate the tumor microenvironment, ICIs response, and play a tumor suppressor role in HCC. We consider that microRNAs (miRNAs) also play a crucial bridging role between AL365361.1 and NLRP3. Thereafter, target gene prediction software was used to find the possible hub between them. The sunburst diagram shows 51 miRNAs may target NLRP3, 61 miRNAs may be targeted by AL365361.1, and 6 common miRNAs among them (Figure 11B). There were 77 and 209 miRNAs co-expressed with TILs and ICIs related miRNAs, respectively (|R| > 0.2 and *p* < 0.001). Based on the online Venn diagram tool, we screened two common miRNAs from miRNA-NLRP3, AL365361.1-miRNA, TILs-, and ICIs-related miRNAs (Figure 11C). The two common genes were hsa-miR-17-5p and hsa-miR-93-5p. The correlation heatmap between the two common genes and the main immune checkpoint genes showed that hsa-miR-17-5p and hsa-miR-93-5p might play a carcinogenic role in HCC (Figure 11D). In addition, by visualizing miRNAs targeting NLRP3 (Figure 11E), we found a significant interaction between hsa-miR-17-5p and NLRP3. Meanwhile, we performed qRT-PCR to examine the expression level of hsa-miR-17-5p in HCC tissues and adjacent normal tissues from the HCC patients. The result showed that the hsa-miR-17-5p expression in HCC tissues is significantly higher than that in adjacent normal tissues (Figure 11F). The correlation of three molecules between the expression levels in 10 pairs of HCC patients’ tissues (Figure 11G–I) was analyzed, respectively. The results showed that AL365361.1 was negatively correlated with hsa-miR-17-5p, NLRP3 was negatively correlated with hsa-miR-17-5p, and AL365361.1 was positively correlated with NLRP3. These are consistent with our predictions. Therefore, we hypothesized that the AL365361.1/hsa-miR-17-5p/NLRP3 axis may be related to the ICIs sensitivity of HCC patients, which is the main reason for the difference in immune response and OS between the two risk groups.

## 4. Discussion

Patients with hepatocellular carcinoma have a poor prognosis, most of them lost the opportunity for surgery when they are in the advanced stage [33]. Immunotherapy combined with targeted therapy brings hope to these patients, but it is difficult to identify sensitive populations and patients prone to drug resistance now [14]. LncRNAs have been reported to be used as biomarkers to predict the efficacy of chemotherapy, targeted therapy, and immunotherapy, but the potential clinical value and regulatory mechanism of crLncRNAs in HCC have not been studied [34,35]. There, we hope to provide a sensitive biomarker to predict the prognosis of HCC patients and immunotherapy efficacy, which will guide the selection of ICIs and targeted drugs by constructing the prognostic model of crLncRNAs, thus serving as a basis for formulating appropriate clinical strategies. The preliminary exploration of the potential mechanism of 9-crLncRNA in dividing the ICIs-sensitive population also provides a research direction for basic research.

Recently, researchers have verified that cuproptosis-related genes could assess prognosis in patients with clear cell renal cell carcinoma [36], melanoma [37], and hepatocellular carcinoma. Compared with the recently established crLncRNA prognostic model in HCC, our model, which consists of nine crLncRNAs (AC026412.3, AC026356.1, MCM3AP-AS1, AL031985.3, AL117336.2, AL365361.1, SLC6A1-AS1, MIR548XHG, AC011462.4), has higher specificity and sensitivity (1-, 3-, and 5-year AUC of 0.828, 0.781, and 0.779, respectively) [38,39]. Patients are divided into high- and low-risk groups based on the risk score, and the OS in each group can be predicted. The low expressions of AL365361.1 and SLC6A1-AS1 in the high-risk group may play an anti-tumor role, while the other seven crLncRNAs have the opposite effect. It has been reported that AC026412.3 can predict patient survival in hepatocellular carcinoma and is associated with immune invasion and the tumor microenvironment [40]. MCM3AP-AS1 promotes hepatocellular carcinoma growth by targeting the miR-194-5p/FOXA1 axis [41]. AL031985.3 is associated with ferroptosis-related hepatocellular carcinoma and predicts patient survival and immunotherapy response [42]. It is possible to predict early recurrence of hepatocellular carcinoma after curative resection with AL365361.1 [43]. Although some of the nine crLncRNAs have been reported to be able to predict the prognosis of HCC by other studies, these predictive LncRNAs were not used to jointly construct the prognosis model. Therefore, we have constructed the model to predict the prognosis and ICIs treatment efficacy of HCC patients, and explored the potential function of the nine crLncRNAs in the model. This is the first time nine crLncRNAs have been used as a biomarker to predict the efficacy of ICIs therapy in patients with HCC.

Our results showed that patients in the low-risk group, divided according to their risk score, had better OS, better treatment efficacy of ICIs, and were more likely to benefit from ICIs treatment. The reason may be that immune cell infiltration in the low-risk group enhanced their immune reactivity. The nine crLncRNAs are enriched in the tumor immune pathway and may be related to tumor immunity. The results also explain why our model can divide the population with different immunotherapy efficacies. Some studies have screened sensitive targeted therapeutic drugs by GDSC analysis and verified the effect of sensitive drugs on tumor progression through experiments [44,45]. Therefore, the genomic resistance analysis using Cancer Drug Sensitivity Genomics (GDSC) IC50 drug data is feasible for the patients in the high-risk group with low immunotherapy sensitivity. The results show that patients in the high-risk group may have higher sensitivity to VNLG/124, sunitinib, and linifanib and better clinical benefits. Therefore, our prognostic model can predict the OS of HCC patients, evaluate the sensitivity of ICIs, and choose the appropriate treatment for different groups.

Interestingly, AL365361.1 is the core molecule of crLncRNA in the model and may be the main factor affecting the difference in ICIs response between low- and high-risk populations. We found that AL365361.1 may affect the interaction between immune checkpoints, probably mainly CD28 and CD40LG. CD28 is essential for maintaining immune homeostasis and ensuring T cell survival [46,47]. CD28 may enhance effector T function and block the inhibitory function of Treg cells [48]. CD40/CD40L molecular pairs can mediate bidirectional signaling between T cells and APC, through reverse signaling that leads to activation and differentiation of APC as well as positive signaling that leads to activation and differentiation of T/B cells [49,50]. Our results showed that AL365361.1 expression in HCC tissue is significantly lower than in adjacent normal tissue, as expected. This implies that higher expression of AL365361.1 by affecting CD28 and CD40LG leads to a higher immune response in the low-risk group. Therefore, we further explored and verified the regulatory network of AL365361.1 in HCC tissues. It is exciting to find that the AL365361.1/hsa-miR-17-5p/NLRP3 axis may inhibit HCC progression and may be related to the ICIs sensitivity of HCC patients, which is the main reason for the difference in immune response and OS between the two risk groups. Moreover, it has been reported that hsa-miR-17-5p promotes the progression of HCC and is related to sorafenib resistance [51]. Sorafenib is a receptor tyrosine kinase inhibitor that can significantly suppress HCC growth [52]. Previous studies did not pay attention to the effect of AL365361.1 on immunotherapy efficacy in HCC, or the effect of hsa-miR-17-5p and its target NLRP3, on drug resistance. By intervening with the AL365361.1/hsa-miR-17-5p/NLRP3 axis, it is possible to increase the objective response rate of ICIs, while at the same time providing new treatment options for ICIs-insensitive populations.

Although CD8+ cytotoxic T cells play key roles in eliminating tumor cells, they often experience exhaustion late in development and are unable to control tumor progression [53]. NLRP3 may be a bidirectional valve for the interaction between inflammation and EMT [54,55]. Activation of NLRP3 can drive an inflammatory response, while inhibition of NLRP3 may aggravate immune exhaustion in patients with advanced HCC, which is not conducive to ICIs treatment and leads to poor efficacy of ICIs treatment in patients in the high-risk group [56,57,58]. Genomic mutations that influence clinical responses to therapy can guide the selection of targeted drugs. NLRP3 may be a valve for reversing treatment outcomes in high-risk patients, and we screened sensitive drugs (VNLG/124, sunitinib, and linifanib) for NLRP3 mutation. VNLG/124 was reported to be able to enhance the anticancer activity of breast and prostate cancer cells in vitro [59]. TACE plus sunitinib can prolong the survival of patients with unresectable HCC [60]. Sunitinib can increase the anti-tumor immune response by inhibiting the hepatocyte growth factor (HGF) and vascular endothelial-derived growth factor (VEGF) signaling pathways [61,62]. Linifanib achieved OS similar to sorafenib in advanced HCC patients with sorafenib intolerance [63]. These clearly verify the efficacy of VNLG/124, sunitinib, and linifanib in the treatment of HCC, and when NLRP3 is mutated, the sensitivity may be enhanced in the high-risk group of patients with low NLRP3 expression, thus bringing better clinical benefits.

The research may predict patient survival and guide the selection of immunotherapy and targeted therapy for patients in high- and low-risk groups. Furthermore, our study could evaluate the efficacy of immunotherapy, and provide a feasible approach for identifying significant challenges in treating sensitive populations. Inevitably, our study has limitations, and more prospective data are needed to verify the clinical application value of our 9-crLncRNA prognostic model in guiding treatment selection.

## 5. Conclusions

The study found that the prognostic model based on 9-crLncRNA had excellent specificity and sensitivity and was able to predict the prognosis of HCC patients and identify ICIs sensitive populations. Additionally, tumor immunity may be connected to tumor cuproptosis. According to the crLncRNA model, patients in the low-risk group may have higher ICIs sensitivity and benefit from ICIs therapy, and the crLncRNA AL365361.1 may affect the immune response to ICIs through the AL365361.1/hsa-miR-17-5p/NLRP3 axis. It is possible to improve the treatment outcomes by using NLRP3 mutation-sensitive drugs (VNLG/124, sunitinib, and linifanib) in the high-risk group of patients who are not sensitive to ICIs and have low expression of NLRP3. In a word, our study may provide new insights into the development of appropriate clinical strategies.

## Figures and Tables

**Figure 1 cancers-15-00544-f001:**
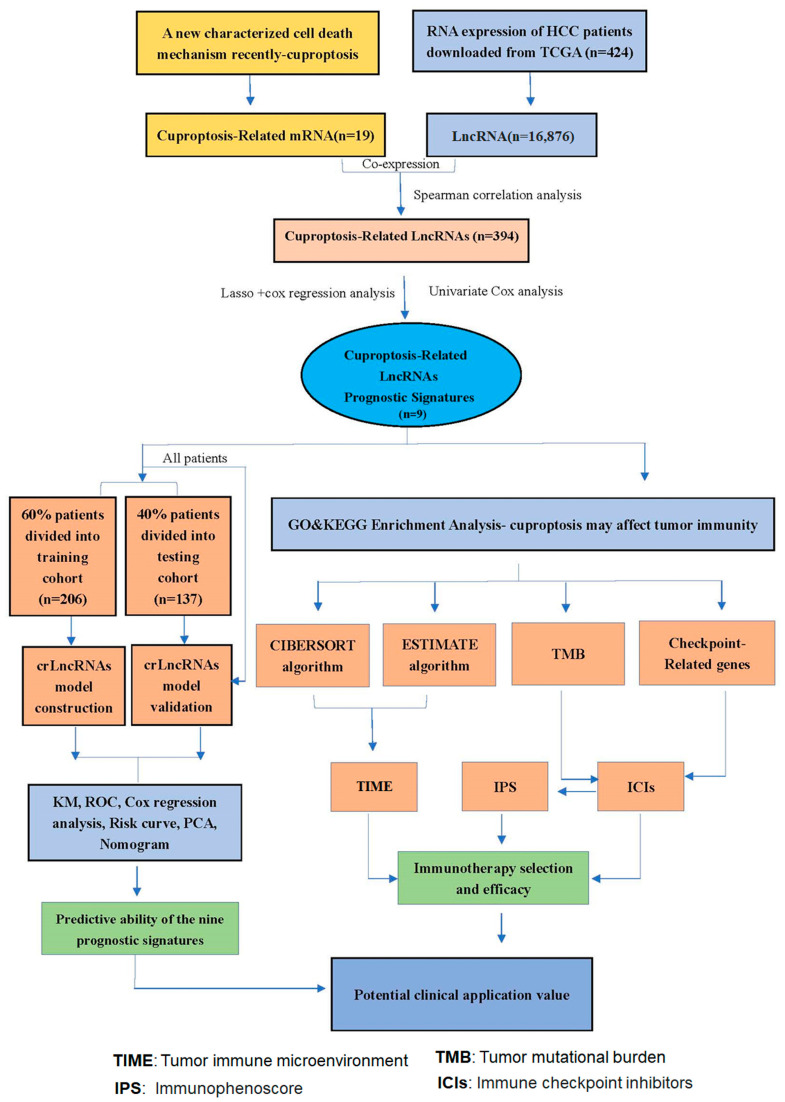
Flow chart of this study. A total of 343 HCC patients with complete survival information in the TCGA database were divided into two cohorts, i.e., training and testing. Using the expression data of 147 prognostic crLncRNA genes in the training cohort, the 9-crLncRNA signatures based on LASSO and COX regression analyses were obtained, and the optimal penalty parameter (λ) of the LASSO model was constructed. The KM curves, ROC curves, PCA analysis, DCA curves, and nomogram were applied to evaluate the accuracy and reliability of the 9-crLncRNA prognostic model. Subsequently, a series of analyses, including ssGSEA, KEGG, GO, GSVA, immune-related, immunecheckpoint-related genes, somatic mutations, and drug sensitivity analyses, were applied to explore the potential of classifying the population as sensitive to treatment and its mechanism.

**Figure 2 cancers-15-00544-f002:**
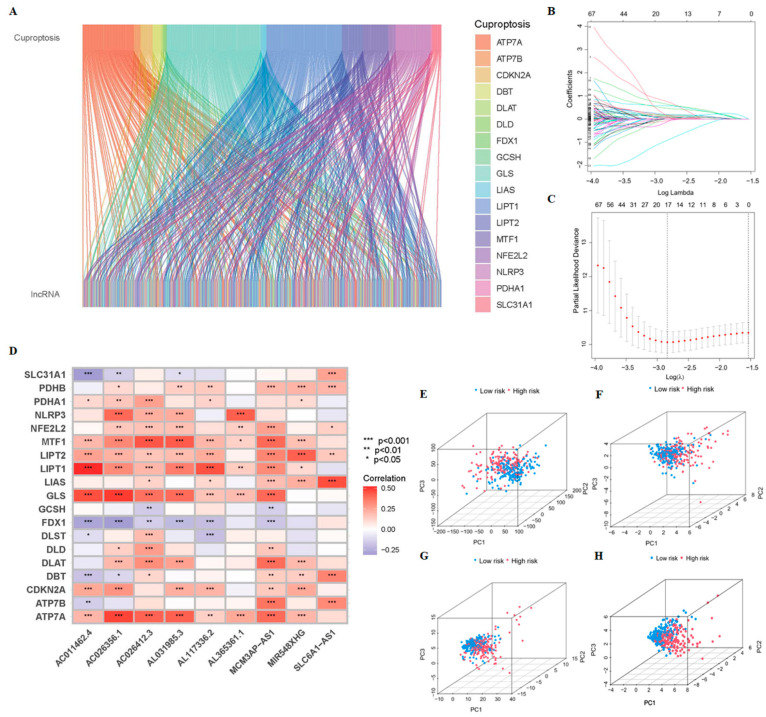
Identification of crLncRNAs in HCC patients, the prognostic model of crLncRNAs and its prediction potential. (**A**) The co-expression network of mRNAs and LncRNAs associated with cuproptosis, which was visualized by Sankey diagram according to Spearman correlation analysis (|R| > 0.4, *p* < 0.001). (**B**) A cross-validation to adjust the parameter selection in the LASSO Cox regression model. (**C**) The LASSO coefficient curve of 9 crLncRNAs. (**D**) The heatmap of correlation analysis of cuproptosis-related genes and the 9-crLncRNA signatures. * *p* < 0.05, ** *p* < 0.01, and *** *p* < 0.001. Principal components analysis between low- and high-risk groups based on the expressions of (**E**) entire genes (**F**) 19 cuproptosis-related genes (**G**)147-crLncRNA, and (**H**) 9-crLncRNA.

**Figure 3 cancers-15-00544-f003:**
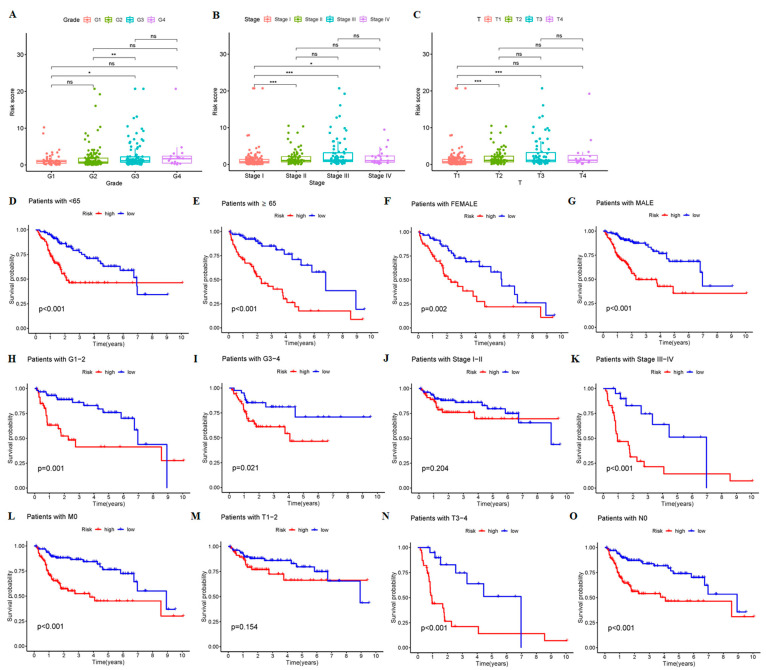
Validation of the 9-crLncRNA prognostic model by clinical characteristics. Based on the median risk score, the entire sample was divided into high-risk group (165 cases) and low-risk group (178 cases). The differences in (**A**) grade, (**B**) stage, and (**C**) T stage were analyzed in HCC patients. * *p* < 0.05, ** *p* < 0.01, and *** *p* < 0.001, and ns means no significance. The K-M plotters of clinical characteristics: (**D**) <65, (**E**) ≥65, (**F**) FEMALE, (**G**) MALE, (**H**) G1-G2, (**I**) G3-G4, (**J**) stage I–II, (**K**) stage III–IV, (**L**) M0, (**M**) T1-T2, (**N**) T3-T4, and (**O**) N0. *p* < 0.05 was considered statistically significant.

**Figure 4 cancers-15-00544-f004:**
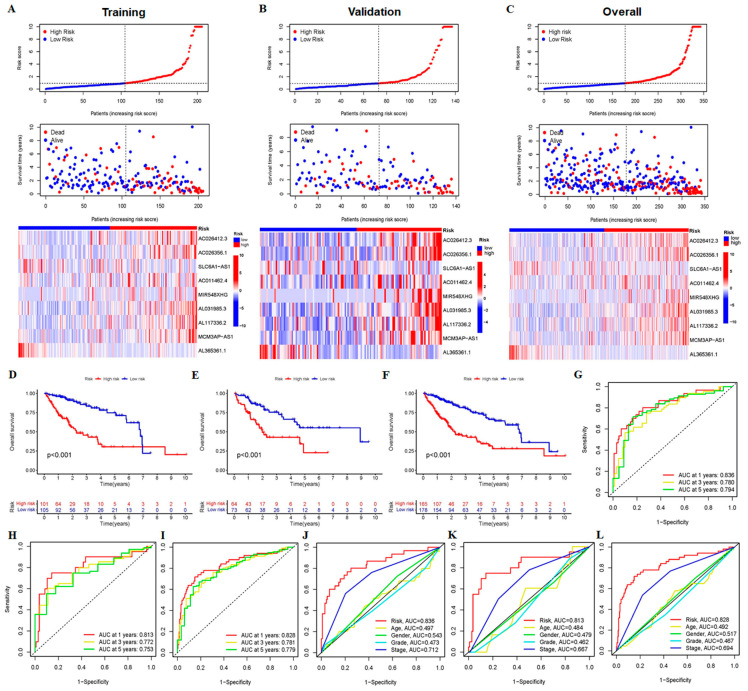
Validation of the 9-crLncRNA prognostic model in the training, validation and entire groups. The training group was further divided into the high-risk group (101 cases) and the low-risk group (105 cases) based on the median risk score. The validation sample was then divided into the high-risk group (64 cases) and the low-risk group (73 cases). Each sub-graph of Figure 4 shows the risk score curves, survival status distribution maps, the 9-crLncRNA expression heatmaps (**A**–**C**), Kaplan–Meier survival curves (**D**–**F**), the ROC curves of the overall survival at years 1, 3, and 5 (**G**–**I**), and the ROC curves of the risk score and other relevant clinical characteristics of the 9-crLncRNA (**J**–**L**) for the training, validation and entire groups, respectively. *p* < 0.05 was considered statistically significant.

**Figure 5 cancers-15-00544-f005:**
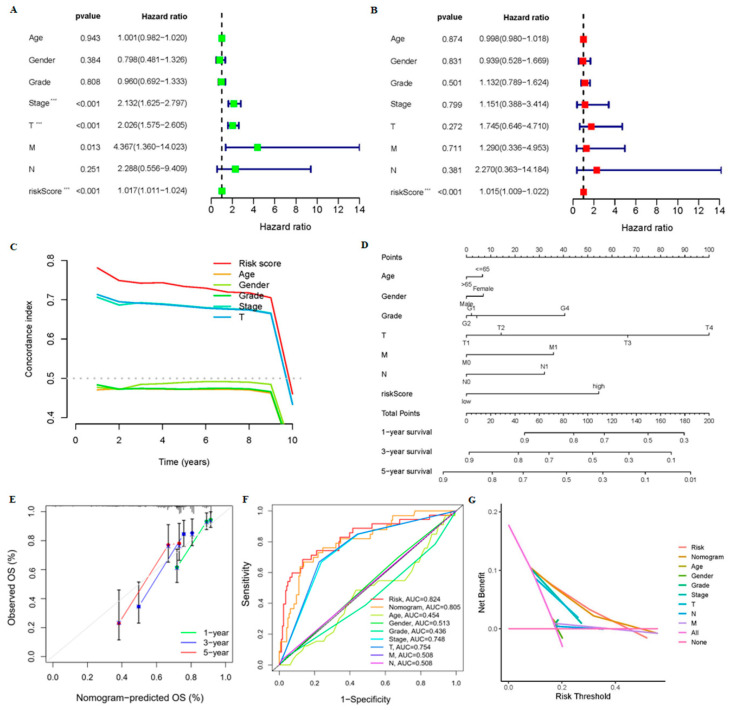
Evaluation of the predictive ability of the 9-crLncrNA prognostic model. The univariate (**A**) and multivariate (**B**) Cox regression analyses were performed to evaluate the independent predictive potential of OS of the risk score and relevant clinical characteristics. *** *p* < 0.001. (**C**) The C-index was used to evaluate the predictive power of the model. (**D**) The nomogram was used for the prediction of 1-, 3-, and 5-year survival. (**E**) The calibration curves were used to examine the capability of predicting the OS at 1, 3, and 5 years. (**F**) The multi-indicator ROC curve and (**G**) the DCA curve were used to evaluate the predictive ability of the nomogram and risk score.

**Figure 6 cancers-15-00544-f006:**
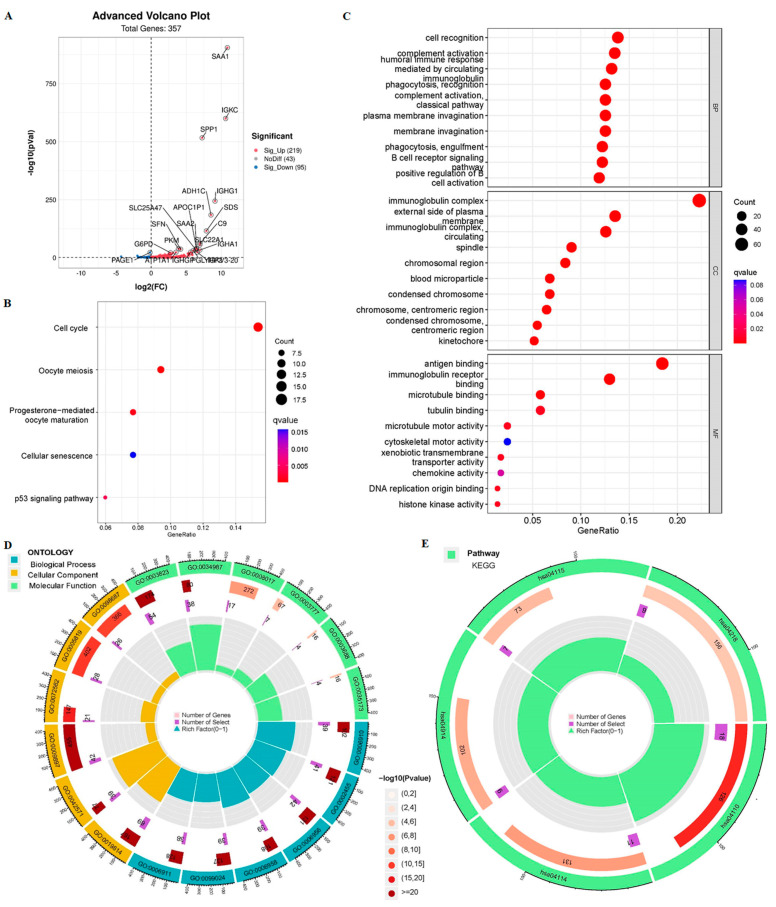
The GO biofunction and KEGG pathway enrichment analysis of two risk groups based on the 9-crLncRNA. (**A**) The volcano map reflects the 357 differentially expressed genes between two risk groups (log2|FC| > 1, *p* < 0.05). Green, red, and gray represent downregulated, upregulated, and no difference genes, respectively. Bubble graphs for KEGG pathways (**B**) and GO enrichment (**C**) Circle diagrams of significant GO functional items (**D**) and significant KEGG pathways (**E**). The latter contains the name of the dataset, the number of genes in the dataset, and the proportion of crLncRNAs in the pathway. The outer ring is the name of the dataset, the inner circles are the number of genes in the dataset and the proportion of crLncRNAs in the pathway. *p* < 0.05 was considered statistically significant.

**Figure 7 cancers-15-00544-f007:**
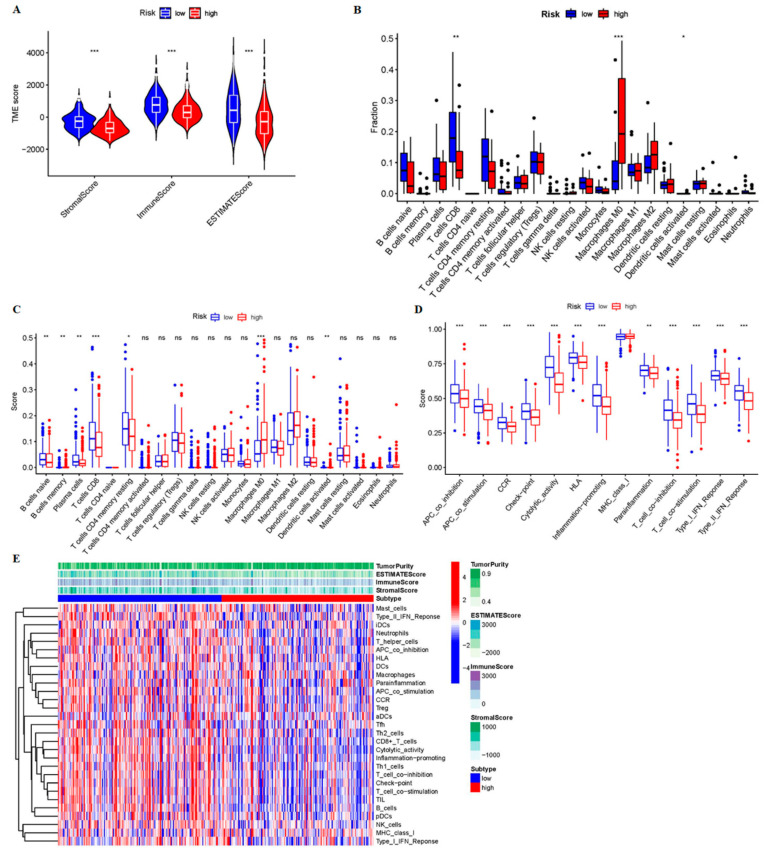
Immune profiles between different risk groups. (**A**) The boxplots of the stroma, immune, and estimate scores of the two groups, and the Wilcoxon test is used for comparison. (**B**) The comparison of immune cell subtypes in the low- and high-risk groups. The differences of (**C**) immune cell infiltration and (**D**) immune function between the two risk groups according to ssGSEA. (**E**) The heatmap summarized stroma, immune, estimate scores, tumor purity, immune cell infiltration, and immune function between the two groups. * *p* < 0.05, ** *p* < 0.01, and *** *p* < 0.001, ns means no significance.

**Figure 8 cancers-15-00544-f008:**
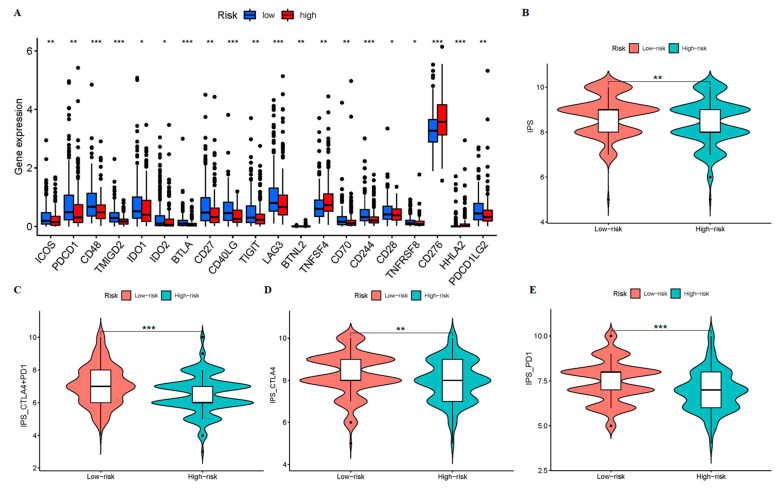
The comparison of the expressions of immune checkpoint genes and sensitivity to immune checkpoint inhibitors between high- and low-risk groups. (**A**) The boxplots for comparing the immune checkpoints genes between the two risk groups. The violin figures are for comparing the two risk groups of the treatment for using (**B**) none of CTLA4 or PD1, (**C**) CTLA4 + PD1, (**D**) CTLA4 alone, and (**E**) PD1 alone. * *p* < 0.05, ** *p* < 0.01, and *** *p* < 0.001.

**Figure 9 cancers-15-00544-f009:**
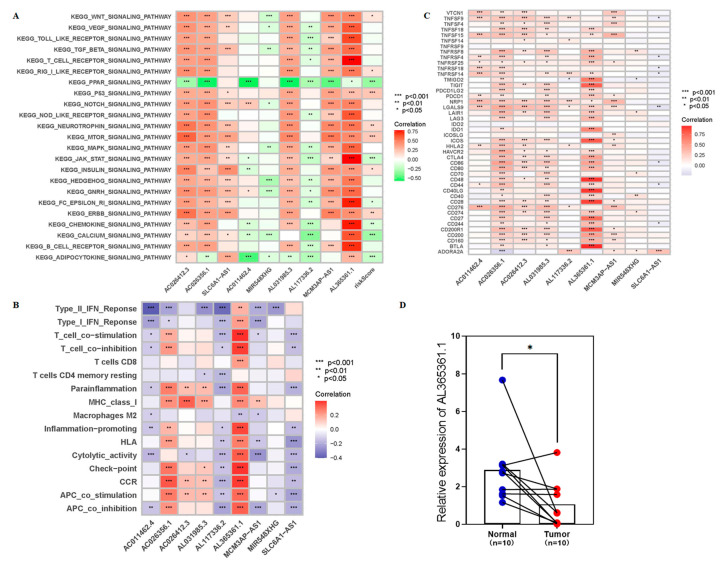
The connection between 9-crLncrNA prognostic signature and the immune landscape. (**A**) Correlation heatmap of KEGG pathways by GSVA. (**B**) The 9-crLncrNA correlates with 22 immune cells and immune-related functions on a heatmap. (**C**) The heatmap of the correlation between the 9-crLncrNA and immune checkpoint genes. (**D**) The differential analysis plots show the expression of AL365361.1 in HCC tissue and adjacent normal tissue from HCC patients. * *p* < 0.05, ** *p* < 0.01, and *** *p* < 0.001.

**Figure 10 cancers-15-00544-f010:**
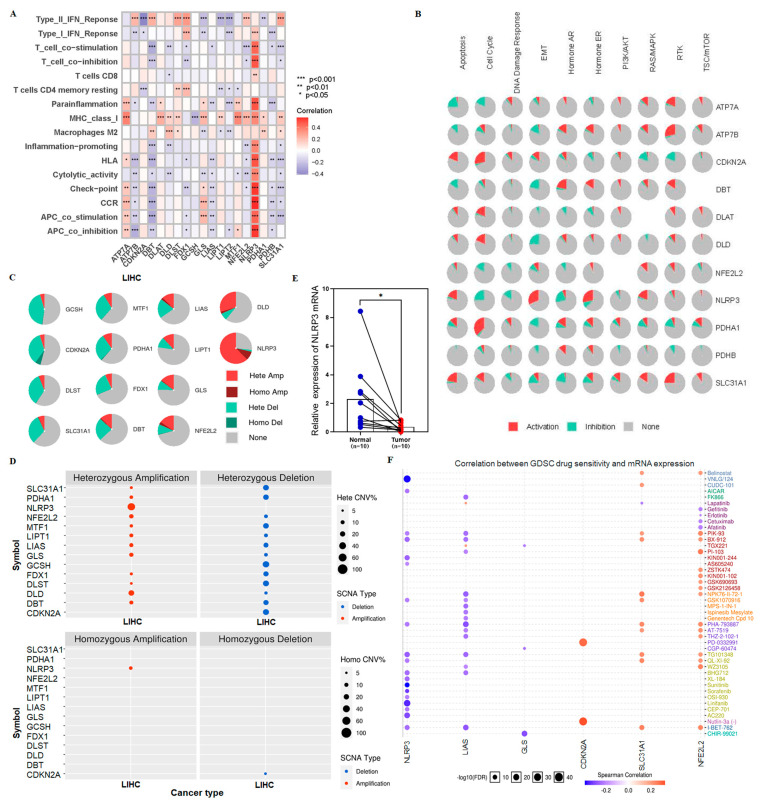
The analysis of drug sensitivity in high-risk group. (**A**) The heatmap of correlation between cuproptosis-related genes and 22 immune cells and immune-related functions. (**B**) The pathways pie chart of cuproptosis-related genes, with red representing activation and green representing inhibition. (**C**) The pie charts of cuproptosis-related genes with CNV in HCC. (**D**) The CNV status of the cuproptosis-related genes in heterozygous and homozygous HCC patients was shown above and below the figure, respectively. (**E**) The differential analysis plots show the expression of NLRP3 in HCC tissue and adjacent normal tissue from HCC patients. (**F**) The correlation between cuproptosis-related gene expression and sensitive targeted therapy drugs obtained by the spearman correlation analysis. The red represents positive correlation, indicating that the gene is highly expressed and resistant to drugs, whereas the blue represents the drug sensitivity of the gene. * *p* < 0.05, ** *p* < 0.01, and *** *p* < 0.001.

**Figure 11 cancers-15-00544-f011:**
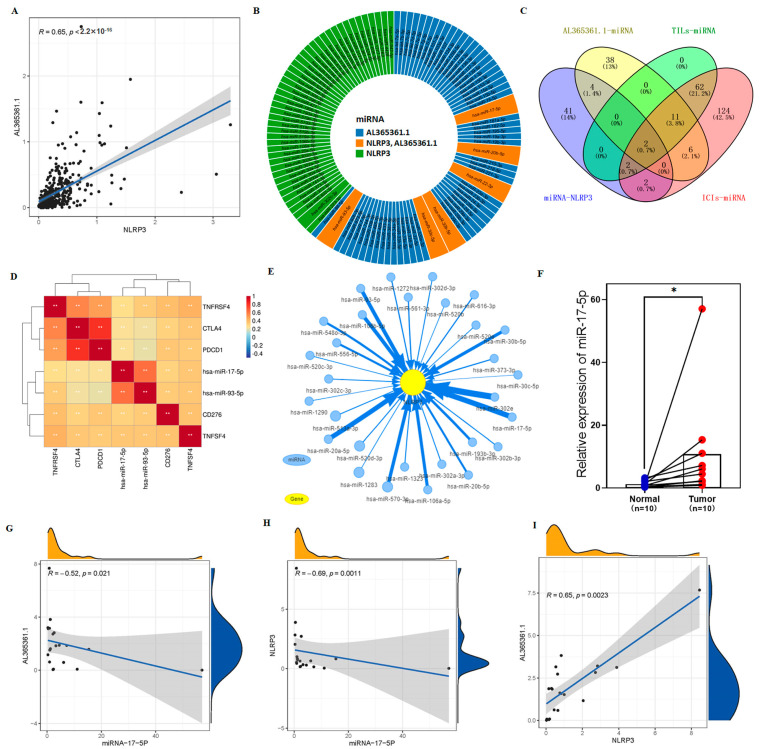
The regulatory network between AL365361.1 and NLRP3. (**A**) The scatter plot of correlation showing the correlation of the expression levels between AL365361.1 and NLRP3. (**B**) The sunburst diagram shows predicted microRNA targets of NLRP3 and AL365361.1. (**C**) The Venn diagram containing four lists of miRNAs. (**D**) The correlation heatmap between the 2 common miRNAs and the main immune checkpoint genes. (**E**) The network of the potential regulation of miRNAs to NLRP3. (**F**) The differential analysis plots show the expression of miR-17-5p in HCC tissue and adjacent normal tissue from HCC patients. The scatter plot of correlation shows the correlation of the expression levels between AL365361.1 and hsa-miR-17-5p (**G**), NLRP3 and hsa-miR-17-5p (**H**), AL365361.1 and NLRP3 (**I**). * *p* < 0.05, ** *p* < 0.01. *p* < 0.05 was considered statistically significant.

## Data Availability

Any data and R script in this study can be obtained from the corresponding author upon reasonable request. The final manuscript was read and approved by all authors. In this study, the RNA—Seq data and corresponding clinical characteristics were obtained from the TCGA database (https://portal.gdc.cancer.gov/, accessed on 31 May 2022). The absolute abundance of immune and stromal cell expression profiles were obtained from CIBERSORT (http://CIBERSORT.stanford.edu/, accessed on 31 May 2022). The IPS of HCC patients was obtained from the Cancer Immunome Atlas (https://tcia.at/, accessed on 31 May 2022). The gene set and IC50 drug data was obtained from GDSC (https://www.cancerrxgene.org/, accessed on 22 June 2022).

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
