# Peer review of "Revealing Prognostic and Immunotherapy-Sensitive Characteristics of a Novel Cuproptosis-Related LncRNA Model in Hepatocellular Carcinoma Patients by Genomic Analysis"

_cancers, 2023, doi:10.3390/cancers15020544_

Round 1

Reviewer 1 Report

The authors put the information with a maximum statistical jugglery to prove or disapprove their theory, which is just published this year in Nov, 2022 by some other groups (See Link: https://pubmed.ncbi.nlm.nih.gov/36452343/), This reviewers fails to understand the clinical relevance  of this data and how does it fit to the HHC world, that ultimately goes o Liver Transplantation at its most. Besides this I have some specific comments as below:

1.      First, the authors must make the reviewers understand why they chose the closely alike published work and what is the rationale to go forward. To this reviewer. It is a kind of repetition. I only further review this article after Editors discretion to move forward. Good luck.

Author Response

Response to Reviewer 1 Comments

Comments and Suggestions for Authors

The authors put the information with a maximum statistical jugglery to prove or disapprove their theory, which is just published this year in Nov, 2022 by some other groups (See Link: https://pubmed.ncbi.nlm.nih.gov/36452343/), This reviewers fails to understand the clinical relevance of this data and how does it fit to the HHC world, that ultimately goes o Liver Transplantation at its most. Besides this I have some specific comments as below:

Response: Thank you for your valuable and thoughtful comments. In the research, we downloaded RNA-Seq data and corresponding clinical characteristics from the Cancer Genome Atlas (TCGA) database (424 HCC samples, including 50 normal samples and 374 tumor samples). By analyzing the data, we constructed the 9-CrLncRNA prognostic model, which exert effective predictive capability of HCC patients. The TCGA, a representative database, is the theoretical basis of this study. So we hope that it can fit to the HCC. Moreover, the Liver transplantation you mentioned is indeed an exciting and challenging research. It is promising for hepatocellular carcinoma patients, but the indications are strict. Some groups tried to establish models that could plan the timing of transplantation or predict the recurrence of hepatocellular carcinoma after transplantation by evaluating pretransplant cytokine profiles, tumor load and other indicators prior to transplantation. At the same time, some researchers paid attention to the construction of liver transplantation risk prediction model, but it is difficult to build a satisfactory transplantation risk prediction model (For example, PMID: 35211739; PMID: 35124188) due to large differences in etiology, transplantation needs and individuals, which also requires matching clinical transplantation data. However, it is pity that we had not pay much more attention to the liver transplanation, our original intention in constructing the model was to predict patient survival and identify sensitive populations of immunotherapy patients. Therefore, the model is not recommended to predict the risk of HCC recurrence in liver transplantation. But the construction of liver transplantation risk prediction model is a promising direction of research to further improve HCC prognosis. We hope the response is acceptable.

  1. First, the authors must make the reviewers understand why they chose the closely alike published work and what is the rationale to go forward. To this reviewer. It is a kind of repetition. I only further review this article after Editors discretion to move forward. Good luck.

Response: Thank you for your very insightful comments. Actually, a team established a prognostic model with a good predictive efficacy, and the AUC of 1, 3 and 5 years being 0.717, 0.633 and 0.607, respectively. The paper was published in November 2022. Interestingly, we also accomplished the construction of a cuproptosis prognostic model in the meantime, which also shows the significance of this study. The data suggested that the AUC of our model was 0.828, 0.721 and 0.779 in 1, 3 and 5 years, respectively, showing higher specificity and sensitivity than previous studies. Moreover, the AUC was positively correlated with the prediction efficiency, especially in the data analysis phase. For example, the test kit of 7 microRNAs for the diagnosis of Hepatitis B virus-associated hepatocellular carcinoma (see Link: https://pubmed.ncbi.nlm.nih.gov/22105822/), which was developed by Academician Jia Fan's team, showed the high stability of prediction efficiency in late clinical validation. Its AUC is 0.842. Besides, our study further analyzed the potential clinical value of the molecules that constitute the model in HCC and suggested therapeutic strategies for different at-risk populations. Finally, we investigated the molecules that may play a key role in identifying immunotherapy-sensitive populations, and proposed the potential mechanisms that may influence the immune response. We attempted to step-by-step dig deeper and apply the model to solve clinical problems. Therefore, we think that our study is different from the article published in November 2022, and they complement each other which can better suggest the important implications of the cuproptosis-related lncRNA model for the prognosis of HCC. We have cited the paper at the suitable position according to your kindly reminder. We truly hope this explanation is acceptable to you.

Reviewer 2 Report

In this manuscript, the authors (as mentioned in the abstract) would like to provide results that show cuproptosis-related LnRNAs may have a predicting role in the survival of HCC patients and improve the current problem of immunotherapy. First of all, the main objectives sound very interesting. However, I have the following concern about the manuscript.

Minor suggestions:

1) As authors are working on HCC, they mentioned in the abstract, "It’s a main problem to improve the objective response rate of immunotherapy in combination with targeted therapy in HCC treatment." This is obscure. They should provide more clear information about HCC immunotherapy and its challenges. Especially, authors should mention nivolumab (PD-1 inhibitor), which received accelarated approval for HCC but was recently withdrawn by FDA. 2-3 sentences with relevant references in the introduction are fine.

2) Page 6 (Ln 216): The authors did not mention how they found this formula.

3) Page 7: Authors started the manuscript with goals they would like to find a solution for immunotherapy, but the majority of genes they analyzed are not specific for immune response.

4) I found an almost similar published paper (PMID: 36452343) that was not cited in the manuscript (CrLnRNA in HCC).

Major suggestion:

As I mentioned above, the objectives sound very interesting at first look. However, as I proceeded with the reading, I found the direction of the manuscript was gradually changing. This makes the manuscript challenging to follow. The conclusion is not as expected.

4) Pages 16-17. The authors focused on NLRP3. As I have a look at TCGA, I found this gene expression is very low in both HCC patients and normal. Are there any significant differences between the expression of NLRP3 in HCC tumors and normal liver tissue?

5) As the authors mentioned (Page 16), NLRP3 has associated T cells and immune checkpoints.  However, authors have focused on targeted therapy drugs such as sunitinib and linifanib, tyrosine kinase inhibitors, not checkpoint inhibitors. What is the rationale behind the selection of these drugs?l

6) If the authors would like to provide evidence that CrLnRNAs are important in immunotherapy, the research design should be so that to confirm this objective. Otherwise, the manuscript should be rewritten, or part of the analysis should be based on sensitivity and resistance to immunotherapy drugs like nivolumab.

7) Is this pathway and CrLnRNA affect checkpoint inhibitors such as nivolumab or atezolizumab? Can we use Cr-LnRNA to predict the response of nivolumab in HCC patients?

Author Response

Response to Reviewer 2 Comments

Comments and Suggestions for Authors

In this manuscript, the authors (as mentioned in the abstract) would like to provide results that show cuproptosis-related LnRNAs may have a predicting role in the survival of HCC patients and improve the current problem of immunotherapy. First of all, the main objectives sound very interesting. However, I have the following concern about the manuscript.

Minor suggestions:

1) As authors are working on HCC, they mentioned in the abstract, "It’s a main problem to improve the objective response rate of immunotherapy in combination with targeted therapy in HCC treatment." This is obscure. They should provide more clear information about HCC immunotherapy and its challenges. Especially, authors should mention nivolumab (PD-1 inhibitor), which received accelarated approval for HCC but was recently withdrawn by FDA. 2-3 sentences with relevant references in the introduction are fine.

Response: Thank you for your thoughtful comments. We have refined the presentation in the abstract and added the ICIs relevant references in the introduction (Line 24-26 and Line 54-59) according to your suggestion. Actually, we also paid attention to the withdrawal of nivolumab monotherapy for HCC. However, there are also studies on the clinical efficacy and safety of nivolumab plus ipilimumab. Moreover, immunotherapy has greatly changed the treatment options for HCC and brought a turning point for extending the survival time of HCC patients. Therefore, we attempted to construct a crLncRNA model that could identify immunotherapy-sensitive populations, so that some patients could gain better clinical benefits through immunotherapy. We hope the revision is acceptable.

2) Page 6 (Ln 216): The authors did not mention how they found this formula.

Response: Thank you very much for your detailed advice. We had explained how we found this formula at line 110-120, but the specific formula was separately shown in Results section. Maybe it is not convenient to read. Now we have revised the manuscript in section 2.3 according to your kindly reminder.

3) Page 7: Authors started the manuscript with goals they would like to find a solution for immunotherapy, but the majority of genes they analyzed are not specific for immune response.

Response: Thank you for your very insightful comments. We would love to explain why most of the genes analyzed are not specific to immune responses, but rather model-building molecules. Firstly, we tried to build models to predict prognosis and immunotherapy sensitivity in HCC patients. Interestingly, it seems that the model can better predict the prognosis of HCC patients and identify the population sensitive to immunotherapy. Its high predictive performance aroused our interest in deep analysis. Therefore, we further analyzed the molecules which were used to construct the model and performed the immunotherapy sensitivity analysis with immunoresponse-related indicators. This exploratory analysis process also seems reasonable and acceptable. In addition, existing studies about the functions of the molecules in the model are preliminary, which may neglect the understanding of the immune-related functions of the molecules (For example, PMID: 32228580; PMID: 36212138). We hope the response is acceptable.

4) I found an almost similar published paper (PMID: 36452343) that was not cited in the manuscript (CrLnRNA in HCC).

Response: Thank you very much for your kindly reminder. I am sorry we omitted this article. Because our manuscript was accomplished in November, and this paper could not be found at that time. Now we have cited the paper at the suitable position.

Major suggestion:

As I mentioned above, the objectives sound very interesting at first look. However, as I proceeded with the reading, I found the direction of the manuscript was gradually changing. This makes the manuscript challenging to follow. The conclusion is not as expected.

4) Pages 16-17. The authors focused on NLRP3. As I have a look at TCGA, I found this gene expression is very low in both HCC patients and normal. Are there any significant differences between the expression of NLRP3 in HCC tumors and normal liver tissue?

Response: Thank you for your thoughtful comments. We found NLRP3 in the exploratory analysis process. Although its expression level was relatively low compared to many genes, we confirmed the difference of NLRP3 expression between carcinoma and adjacent tissue in TCGA, GEO data, and our own collected clinical specimens (as shown below). We have not employed biological experiments to further confirm its possible important role in hepatocellular carcinoma. However, some investigators have conducted and published the related research results (For example, PMID: 34502191).

 ï¼ˆPlease see the attachment)

5) As the authors mentioned (Page 16), NLRP3 has associated T cells and immune checkpoints. However, authors have focused on targeted therapy drugs such as sunitinib and linifanib, tyrosine kinase inhibitors, not checkpoint inhibitors. What is the rationale behind the selection of these drugs?

Response: Thank you very much for your valuable comments. In the study, we have constructed the 9-crLncRNA model. The data showed that patients in the low-risk group are sensitive to immunotherapy and may obtain better clinical benefits with immune checkpoint inhibitors (ICls) treatment. Moreover, in order to better solve the clinical problems using the model, we tried to solve the treatment problems of high-risk group patients which may not be sensitive to immunotherapy. Our results showed that patients in the high-risk group who are not sensitive to ICls are dominated by NLRP3 mutations. They may not benefit from immunotherapy compared to the patients in low-risk group. It has also been reported that NLRP3 inhibition significantly repressed the expression of immune checkpoints (PD-L1 and LAG3), which is not conducive to ICIs therapy (PMID: 35026655). Subsequently, we employed genomic resistance analysis as shown in Figure 10F, the data showed that sunitinib, and linifanib were sensitive drugs in the inflammatory body NLRP3 mutant population. Therefore, we proposed that the patients in high-risk group should be considered for NLRP3 mutation-sensitive targeted drugs: sunitinib and linifanib.

6) If the authors would like to provide evidence that CrLnRNAs are important in immunotherapy, the research design should be so that to confirm this objective. Otherwise, the manuscript should be rewritten, or part of the analysis should be based on sensitivity and resistance to immunotherapy drugs like nivolumab.

Response: Thank you very much for constructive comments. In order to verify the important role of CrLncRNA in immunotherapy, we conducted exploratory analysis step-by-step. It has been reported that immune checkpoint genes, TIME and TIL are correlated with patient prognosis and sensitivity to immunotherapy. Therefore, we conducted a multidimensional analysis of the important role of CrLncRNA in immunotherapy. Then, IPS scores including treatment regiments in the TCIA database were used to evaluate drug sensitivity of PD-1 and CTLA4 inhibitor in different risk groups. In conclusion, it seems reasonable and acceptable that CrLncRNA model we constructed could be used to distinguish different sensitivity groups and predict the efficacy of immunotherapy. As you mentioned, nivolumabz monotherapy was withdrawn by FDA for HCC patients in 2021. Besides, the online data about drug sensitivity of the model molecules was limited. So, we did not analyze the sensitivity of a particular ICIs. To evaluate the efficacy of ICIs like nivolumab (PD-1 inhibitor), we still need to be cautious and validate with more clinical data. We have revised the manuscript in the Results section (3.6, 3.7, 3.8) according to your suggestions. We tried our best to make it more rigor in the absence of a large amount of clinical data verifying drug sensitivity. We truly hope the revision is acceptable.

7) Is this pathway and CrLnRNA affect checkpoint inhibitors such as nivolumab or atezolizumab? Can we use Cr-LnRNA to predict the response of nivolumab in HCC patients?

Response: Thank you for valuable comments. The main objective of this study is to develop a model for the prognostic analysis of HCC. However, in the process of validating the model, we were pleasantly surprised to find that the model can not only be used to distinguish between high-risk and low-risk groups, but also a good indicator of immunotherapy like ICIs treatment. The patients in the low-risk group maybe benefit from ICIs treatment. Therefore, we tried to explore the possible mechanism and found this AL365361.1/hsa-miR-17-5p/NLRP3 axis. We also verified the expression level and relationship between them in clinical samples. Although this axis may take part in the regulation of drug action theoretically, we should pay more attention to the indicative role of this model for ICIs-sensitive populations. Currently, there is no drug database available for the crLncRNAs mentioned in our model. However, the data showed that this model could be able to predict the reactivity of ICIs according to the association analysis between 9-crLncRNAs and immune indicators. And the specific drug may still need to be further confirmed by clinical data. We hope that this explanation is acceptable.

Round 2

Reviewer 1 Report

Thanks for the explanations and the understanding, and making a thorough changes in the manuscript.

Reviewer 2 Report

Thank you for your response. I found some misspellings (e.g., page 2, line 56). Please, check the manuscript for spelling.